# Conformational Stabilization of Gp41-Mimetic Miniproteins Opens Up New Ways of Inhibiting HIV-1 Fusion

**DOI:** 10.3390/ijms23052794

**Published:** 2022-03-03

**Authors:** Mario Cano-Muñoz, Julie Lucas, Li-Yun Lin, Samuele Cesaro, Christiane Moog, Francisco Conejero-Lara

**Affiliations:** 1Departamento de Química Física, Instituto de Biotecnología y Unidad de Excelencia de Química Aplicada a Biomedicina y Medioambiente (UEQ), Facultad de Ciencias, Universidad de Granada, 18071 Granada, Spain; samuele.cesaro91@gmail.com; 2INSERM U1109, Fédération de Médecine Translationnelle de Strasbourg (FMTS), Université de Strasbourg, 67084 Strasbourg, France; julie.lucas@etu.unistra.fr (J.L.); li-yun.lin@etu.unistra.fr (L.-Y.L.); c.moog@unistra.fr (C.M.)

**Keywords:** fusion inhibitor, calorimetry, coiled-coil, envelope glycoprotein, N-terminal domain, antiviral therapy, gp41

## Abstract

Inhibition of the HIV-1 fusion process constitutes a promising strategy to neutralize the virus at an early stage before it enters the cell. In this process, the envelope glycoprotein (Env) plays a central role by promoting membrane fusion. We previously identified a vulnerability at the flexible C-terminal end of the gp41 C-terminal heptad repeat (CHR) region to inhibition by a single-chain miniprotein (named covNHR-N) that mimics the first half of the gp41 N-terminal heptad repeat (NHR). The miniprotein exhibited low stability, moderate binding to its complementary CHR region, both as an isolated peptide and in native trimeric Envs, and low inhibitory activity against a panel of pseudoviruses. The addition of a disulfide bond stabilizing the miniprotein increased its inhibitory activity, without altering the binding affinity. Here, to further study the effect of conformational stability on binding and inhibitory potency, we additionally stabilized these miniproteins by engineering a second disulfide bond stapling their N-terminal end, The new disulfide-bond strongly stabilizes the protein, increases binding affinity for the CHR target and strongly improves inhibitory activity against several HIV-1 strains. Moreover, high inhibitory activity could be achieved without targeting the preserved hydrophobic pocket motif of gp41. These results may have implications in the discovery of new strategies to inhibit HIV targeting the gp41 CHR region.

## 1. Introduction

The HIV/AIDS pandemic is still very active and continues to be one of the world’s largest pandemics to date with more than 40 million people currently living with HIV still representing a worldwide health issue [1,2]. What is more striking is the upsurge in HIV infections over different populations around the world, such as the outbreak in China’s students, where the number of newly diagnosed college students has seen an annual growth rate ranging from 30 to 50% over the past several years [3]. All this together with the fact that HIV newly infects 1.8 million people each year, makes the development of an HIV vaccine a global health priority [4]. However, almost 40 years after the discovery of HIV as the causative agent of AIDS we still do not have a licensed vaccine. Progress has been hindered by the extensive genetic variability of HIV and our limited understanding of the immune responses required to protect against HIV acquisition [5].

This emphasizes the importance of therapeutics to treat the infection. Despite modern Highly Active Antiretroviral Therapy (HAART) having helped to reduce the number of deaths, the absence of an effective vaccine combined with the growing emergence of multi-resistant HIV variants to several of these drugs urges the development of some new anti-HIV compounds directed against the different stages of the virus life cycle, and in particular against the entry of HIV into the cell [1,6].

In order to enter the target human cell, the virus must fuse its membrane with that of the cell. This fusion process is mediated by the Envelope glycoprotein (Env), a non-covalently associated trimer of heterodimers composed of two glycoprotein subunits, gp120 and gp41 [7]. CD4 receptor and co-receptor (CCR5 or CXCR4) binding to gp120 triggers a series of conformational changes that ultimately cause the adoption of a more energetically favorable conformation called the 6 helix bundle (6HB) formed by the N-terminal heptad repeat (NHR) and C-terminal heptad repeat (CHR) regions of gp41. In this 6HB structure, three CHR regions associate externally over an inner helical coiled-coil NHR trimer in an antiparallel fashion. This energetically favorable interaction between NHR and CHR brings viral and host-cell membranes into close proximity promoting fusion and eventually causing infection. Consequently, compounds that interact with either CHR or NHR interfere with this key process and thereby constitute HIV fusion inhibitors [8,9,10]. For this reason, gp41 has become a very attractive target for the development of potential HIV-1 inhibitors.

Different kinds of fusion inhibitors have been described and classified into two major categories regarding whether they interact with the NHR or CHR regions in gp41. Class-1 inhibitors target the exposed hydrophobic grooves of the NHR helical trimer and comprise a variety of molecules including CHR peptide mimetics, artificial D-peptides, natural products and small-molecule compounds and antibodies [11,12,13,14,15]. On the other hand, class-2 inhibitors target the CHR region and generally encompass NHR peptide mimetics that have traditionally been regarded as low activity anti-HIV compounds. This limited potency may be due to the low solubility and the tendency of NHR peptides to aggregate in solution. These problems can be alleviated by engineered stabilized protein constructs that mimic exposed trimeric NHR grooves. However, they have certain advantages, such as their activity against strains resistant to CHR inhibitors [16,17]. Despite these promising therapeutic approaches, the only FDA-approved fusion inhibitor of AIDS/HIV is T20 (enfuvirtide), a CHR-derived peptide whose clinical use has been limited by its short half-life [18] (proteolysis-sensitive and rapid renal filtration) requiring, therefore, high dosage injections at least twice a day. Moreover, the continuous and expensive treatment generates the appearance of T20-resistant viruses. Nevertheless, compounds that are able to interfere with the formation of the gp41 6HB continue to be very attractive targets for drug design strategies [19,20].

Recently, we have developed several protein molecules called covNHR which consist of a single polypeptide chain with three helical regions that fold as an antiparallel trimeric bundle with a structure highly similar to the NHR gp41 region [21,22]. These proteins can be produced recombinantly by expression in *E. coli* with high yields, without any post-translational modification, are easy to purify, very stable and highly soluble [23]. The NHR binding surface has been described as composed of four different hotspots, namely an N-terminal polar pocket (NTP), a shallow middle pocket (MP), a deep and prominent hydrophobic pocket (HP), which has been widely used as a drug discovery target [14], and finally a C-terminal pocket (CTP) adjacent to the HP [23] (Figure 1C). In a recent study, we designed, produced and characterized two single-chain covNHR miniproteins each encompassing only two consecutive pockets out of the four pockets of the NHR groove [24]. Each miniprotein mimics the N- and the C-terminal half of NHR, respectively, and they were called covNHR-N (harboring only NTP and MP) and covNHR-C (exposing only the HP and the CTP). These miniproteins folded autonomously and represent subdomains of NHR, with very different intrinsic stability. Although both covNHR miniproteins could bind their respective complementary CHR peptides with similar affinity, the covNHR-C protein could not bind its target in soluble prefusion Env spikes and did not show any HIV-1 inhibitory activity in vitro. This is probably due to the HP and CTP binding motifs being engaged in a tryptophan clasp involving the side chains of Trp623 (CTP motif), Trp628 and Trp631 (HP motif) that locks Env in its pre-fusion conformation [25,26]. On the other hand, covNHR-N could bind its target in soluble prefusion Env spikes and showed moderate HIV-1 inhibitory activity in vitro. However, it proved to be quite unstable and required to be stabilized by engineering a disulfide bond connecting one of its two loops to its C-terminal end. The stabilized versions of covNHR-N with one disulfide bond (called covNHR-N-SS) showed a similar affinity towards its complementary CHR peptide and similar capability to bind to soluble prefusion Env spikes. Strikingly, these stabilized variants showed improved inhibitory potency against different HIV strains.

Here, we have furtherly increased the stability of covNHR-N by engineering an additional internal disulfide bond stapling the other end of the molecule to test the hypothesis that an increase in stability can lead to a substantial improvement of the affinity to its complementary CHR peptide accompanied by a subsequent improvement in anti-HIV inhibitory potency. This stabilizing strategy has also been implemented in the complete covNHR parent molecule. Therefore, both protein molecules, called covNHR-N-dSS and covNHR-SS, respectively, (Figure 1) were designed, biophysically characterized and tested for direct Env binding upon several variants. Their capacity to inhibit different HIV-1 pseudoviruses and primary isolates, including T20-resistant strains, was also assessed. The results provide valuable knowledge to the development of protein-based antivirals and reveal new ways to exploit the vulnerability of the gp41 CHR region.

## 2. Results

### 2.1. Design and Stabilization of CovNHR Miniproteins

The initial designs of the parent molecules in this study, covNHR and covNHR-N, contain, respectively, the four and two (NTP and MP) of the pockets described in gp41 (Figure 1 and Appendix A). In order to design the new proteins covNHR-SS and covNHR-N-dSS, we used the crystallographic structure of covNHR in complex with C34 peptide (PDB ID: 6R2G, previously determined by our group [22]) as a template. In our previous study, we had achieved a considerable stabilization of over +20 °C in covNHR-N (T_m_ ≈ 41 °C) by engineering a disulfide bridge that connected the first loop of the miniprotein with its C-terminal end (G33C/R94C mutations) resulting in a new variant called covNHR-N-SS [24]. This “staple” strategy was recreated in this study in order to achieve even further stabilization of the molecules. To accomplish that, Disulfide by Design [27], a web-based tool for disulfide engineering in proteins, was used to predict pairs of residues that will likely form a disulfide bond if mutated to cysteines. We found one possibility for disulfide bond creation connecting different structural elements of the protein by X-Cys mutations. For covNHR-N-dSS, residues Ala2 at the N-terminus and Leu64 in loop 2 fulfilled the strict geometric constraints that disulfide bonds usually require and were also mutated to Cys. Accordingly, covNHR-N-dSS contains mutations A2C/L63C and G33C/R94C. The equivalent disulfide bond connecting the N-terminus and loop 2 was also engineered for covNHR-SS, which in this case contains mutations A2C/L111C (Figure 1).

In order to validate and assess in silico the stability and dynamic behavior of the newly designed miniproteins we conducted all-atom explicit-solvent molecular dynamics (MD) simulations of the free molecules. Root mean square fluctuation (RMSF) values, which are a measurement of the average atomic mobility of the residues in the protein, showed an increase in the stability (i.e., lower values) in the sites where X-Cys mutations were placed (Figure 2A,B). CovNHR-SS showed decreased RMSF values in the vicinity of the mutated sites compared to covNHR (Figure 2A), which helped to decrease the overall residue mobility of the N-terminal end and the second loop, two hotspots for protein instability. A similar scenario occurred with covNHR-N-dSS if we compare its RMSF values with those of the covNHR-N parent molecule (Figure 2B), in this case, the four hotspots in the protein chain, the N- and C-term as well as the first and second loops were stabilized by the addition of the two disulfide bonds. Moreover, the mean RMSF values for each new protein were lower than their respective parent molecule (Figure 2A,B).

Figure 2C,D shows the time evolution of backbone root mean square deviations (RMSD) of the miniproteins. All the molecules reached equilibrium within the first 5 ns of MD simulation. However, the disulfide-stabilized miniproteins reached a more stable plateau and maintained it throughout the entire simulation time while their parent molecules showed a less stable profile. This is also supported by the fact that the mean RMSD values for the newly engineered miniproteins were also lower than those from their parent molecules. These results indicate that the engineered disulfide bonds reduce the overall conformational fluctuations of the new proteins compared to their parent molecules.

### 2.2. Biophysical Characterization of CovNHR Variants

Both disulfide-bonded mutants covNHR-N-dSS and covNHR-SS could be expressed and produced recombinantly in *E. coli* with good yields even higher than those of their respective parent molecule. All purification steps were made in the presence of 10 mM β-mercaptoethanol, and a final oxidation step was carried out by extensive dialysis with buffer without a reducing agent. The formation of the disulfide bonds was confirmed in both mutants using Ellman’s assay (Thermo Fisher, Waltham, MA, USA). The protein purity was assessed by SDS-PAGE, and the identity of each protein variant was confirmed by mass spectrometry analysis.

The structure and stability of the miniproteins were characterized using various biophysical techniques (Figure 3). The two miniproteins showed far-UV circular dichroism (CD) spectra typical of a mostly α-helical structure (Figure 3A,B); covNHR-SS has a similar α-helical structure content to its parent molecule covNHR [22]. On the other hand, covNHR-N-dSS has higher negative ellipticity values than covNHR-N and covNHR-N-SS, its parent molecules [24], showing a higher α-helical structure percentage (72.4% against 64% and 67.1%, respectively). This indicates that the newly formed disulfide bridge is stabilizing the antiparallel helical bundle. At the same time, mixtures between each protein and the Y24L peptide, containing the NTP and MP binding motifs (Figure 1C), showed an increase in negative ellipticity relative to the theoretical ellipticities of the spectra calculated as the sum of the spectra of the free molecules (Figure 3A,B). This indicates the acquisition of the helical conformation for the Y24L peptide as a consequence of binding onto the NHR groove of both miniproteins. The ellipticity increase was similar in both complexes indicating a comparable acquisition of helical structure. This was not the case for covNHR-N, which showed a higher relative ellipticity increase when binding to Y24L peptide, as a consequence of the protein acquiring a more ordered and helical structure when bound to Y24L [24].

Thermal denaturation experiments of covNHR-SS and covNHR-N-dSS indicated strong stabilization of the proteins by the disulfide bonds. The melting temperature (T_m_) of covNHR-N-dSS was −79 °C (Figure 3C), a strong increase of +38 °C compared with covNHR-N without any disulfide bridge and an increase of +18 °C compared with the single disulfide bonded covNHR-N-SS [24]. Regarding covNHR-SS, no unfolding transition was observed even after heating up to 98 °C, confirming its extremely high thermostability. The thermal stability of covNHR-SS was estimated by differential scanning calorimetry (DSC) −124 °C (Appendix A), compared to the 105 °C of the covNHR protein under the same conditions [22]. The denaturation peak showed a sharp drop on the high-temperature side of the peak suggesting thermally induced aggregation similar to what happens to covNHR parent molecule. The thermal stability is thus increased by about 18–20 °C with each disulfide bond engineered in both miniproteins.

Both protein variants are highly soluble at physiological pH and the particle sizes of both proteins were assessed by dynamic light scattering (DLS) (Figure 3D). CovNHR-N-dSS showed an apparent hydrodynamic radius (R_h_) of 1.9 nm and covNHR-SS exhibits a R_h_ of 2.8 nm similar to their respective parent molecules and to their theoretical R_h_. This demonstrates that both variants are monomeric at physiological pH in 50 mM sodium phosphate buffer.

### 2.3. Binding of the CHR Peptide to CovNHR Miniproteins

To characterize in detail the thermodynamics of binding of the miniproteins to the complementary CHR peptide, we performed isothermal titration calorimetry (ITC) analysis (Figure 4 and Appendix A and Table 1) by direct titration of the protein solutions with peptide Y24L, corresponding to gp41 residues 638–661, at the second half of CHR. Previously, we determined the interaction between Y24L and covNHR, which happened to be moderately tight (K_d_ = 90 ± 7 nM at 25 °C) [23]. CovNHR-SS exhibited a similar K_d_ of 116 ± 11 nM at the same temperature. The binding enthalpies and heat capacities are also very similar for these two protein variants (−1.6 vs. −1.7 kJ·K^−1^·mol^−1^, Table 1) [23]. This indicates that the presence of the disulfide bond at the N-terminus of the complete covNHR protein has a small influence on the binding thermodynamics to the CHR peptide and suggests that the presence of a continuous NHR coiled-coil structure already provides strong conformational stability to the N-domain for a competent binding capability at the NTP and MP pockets.

On the other hand, the affinity of covNHR-N for Y24L peptide was determined to be much lower (K_d_ = 790 ± 20 nM) and with more negative binding enthalpy and heat capacity as a result of a considerable entropy penalty associated to structural stabilization of covNHR-N upon interaction with the CHR peptide [24]. This binding entropy cost was not reduced in the singly disulfide-bonded covNHR-N-SS [24], which showed very similar binding parameters although slightly reduced binding heat capacity (Figure 4). In marked contrast, the disulfide bond at the N-terminus in covNHR-N-dSS confers a remarkably higher affinity to Y24L (Table 1 and Figure 4), about a 4.5-fold increase. The thermodynamic magnitudes (Table 1, Figure 4A,B) showed significantly more negative binding enthalpy, partially compensated by a higher entropy cost. These magnitudes indicate a tighter interaction in the complex produced by the new disulfide bond. In fact, the binding affinities and Gibb’s energies are close to those measured for the complete covNHR proteins at low temperatures, although they decrease rapidly with temperature due to a more negative heat capacity change.

### 2.4. Binding to Envelope Proteins

We investigated the influence of stabilization by disulfide bonding on the interaction of covNHR-SS and covNHR-N-dSS with their target gp41 CHR region in various Env proteins, both in uncleaved and cleaved trimeric pre-fusion conformations (Appendix A). ELISA experiments showed that the two miniproteins bind efficiently to all Envs (Figure 5), comparable to each parent molecule [24], except for a slightly enhanced binding of covNHR-N-dSS to gp140 CN54 Env compared to covNHR-N-SS.

Gp41 mimetics encompassing the four NHR pockets, i.e., covNHR and covNHR-SS, displayed higher levels of binding than the miniproteins harboring only two pockets, covNHR-N-SS and covNHR-N-dSS, except for the structurally folded trimeric JRFL Env, in good agreement with their higher affinity for CHR. These and our previous results [24] demonstrate that the C-terminal part of CHR in a prefusion-like Env conformation is accessible to interaction by the covNHR miniproteins encompassing the MP and NTP. However, stabilization by disulfide bonds has a small influence on the capability of each protein to bind the CHR region in pre-fusion stabilized Envs.

### 2.5. HIV-1 Inhibition

The inhibitory activities of the miniproteins against HIV-1 infection in vitro were analyzed using the conventional TZM-bl assay (Figure 6). In these studies, we used different HIV-1 pseudovirus strains: two easy to neutralize pseudoviruses (SF162 and MW956.26) [28], two pseudoviruses with pNL4.3 backbone, (pNL4.3 XCS and pNL4.3 DIM) displaying mutations conferring resistance to T20 [22], and one difficult to neutralize primary isolate (CE1176) [28].

The IC_50_ values were compared with those of the parent covNHR proteins (Table 2). The two proteins containing the four NHR binding pockets, covNHR and covNHR-SS, display similar IC_50_ values in the low nanomolar range meaning that the addition of the N-terminal disulfide bond did not significantly modify the inhibitory capacity of the entire NHR groove mimetic. This is consistent with the fact that both proteins show very similar binding to the CHR target, as shown above.

The inhibitory activity of covNHR-N (containing only two pockets) was very poor as a result of its low affinity for its CHR target [24]. However, conformational stabilization by one disulfide bond in covNHR-N-SS and two disulfide bonds in covNHR-N-dSS increased strongly and progressively the inhibitory activity (Figure 6 and Table 2). These observed activity increments do not appear to be a result of an increase in target binding affinity but rather a consequence of the improved conformational stability of the miniproteins. In fact, the first disulfide bond in covNHR-N-SS did not alter the affinity for the Y24L peptide and the three miniproteins show a similar capacity to bind CHR of prefusion trimeric Envs in ELISA experiments.

Strikingly, covNHR-N-dSS exhibits IC_50_ values of 1–3 tens of nanomolar, only 4- to 10-fold higher than covNHR and covNHR-SS, despite the fact that the former miniprotein does not harbor the HP and CTP pockets and has a consequently a much lower affinity for the CHR region.

Compared to T20, the stabilized miniproteins show consistently higher and broader inhibitory activity against the same virus strains, including T20-resistant strains [24]. Additionally, no cytotoxicity, monitored by microscopic examination, was detected even with the highest concentration of the miniproteins used in the assay.

## 3. Discussion

The NHR region has traditionally been considered as a low stability domain in gp41 with a strong tendency to aggregate, and therefore, NHR-based peptides are poor HIV-1 inhibitors. However, this highly preserved region remains to be an attractive target for fusion inhibition and vaccine design. Different approaches to stabilize an exposed NHR coiled-coil trimer have generally involved inter-helical tethering [29], fusion to foldon domains [9], or partial association with CHR sequences [30]. In these studies, a substantial correlation between coiled-coil stability and inhibitory activity has been repeatedly reported but no clear explanation for this correlation has been provided. We have previously demonstrated that using an innovative design and engineering approach, highly stable mimics of a fully exposed NHR region can be produced in a single-chain form [21,22,23,24,31]. Within the NHR coiled coil, its N-terminal region has also been described as an intrinsically unstable sub-domain compared to that of the C-terminal domain or the entire NHR [24,32]. Here we show how the versatility of our single-chain design allows for simple and effective conformational stabilization by disulfide bond engineering. Each disulfide bond is formed spontaneously in the correct configuration and thermally stabilizes the proteins by about 20 °C, consistently with a strong reduction in the conformational entropy of folding [33].

Despite the additive global stabilizing effect of each disulfide bond, only the addition of the second disulfide bond at the N-terminus has a significant influence on the binding affinity to the CHR peptide. The thermodynamics of binding suggest tighter interactions of the doubly disulfide bonded variant with the CHR peptide, possibly due to a reduced conformational dynamics at the peptide-protein complex interface, as a result of a local stabilization effect produced by the disulfide bond as suggested by the MD simulations. Nevertheless, both disulfide bonds contribute to marked increases in HIV-1 inhibitory activity. Disulfide bond stabilization also produced strong increases in anti-HIV-1 activity in other NHR coiled-coil trimer mimetics [10,34]. However, this high affinity, stability and inhibitory activity has, to the best of our knowledge, never been seen before for a NHR gp41 construct lacking the HP, which has been in the last decades the main focus of attention to interfere with the NHR-CHR interaction of gp41 and thereby inhibit fusion and infection [35,36,37]. For instance, stabilized NHR constructs IZN17, IZN23 and IZN36, all containing the HP but differing in the sequence extension of NHR towards the N-terminus, were reported to have similar inhibitory activities, suggesting that the antiviral activity of this class of chimeric NHR peptides is recapitulated in the HP region [10].

In marked contrast, we show here that the HP is not necessary to achieve a potent and broad inhibitory activity in the tens of nanomolar range for an NHR based construct, such as covNHR-N-dSS. The inhibitory activity is relatively close (only 4- to 10-fold lower) to the activity of the complete NHR mimetic covNHR, as well as other NHR mimetics, with IC_50_ values in the low nM range, even for a difficult to neutralize primary virus isolate (Figure 6). This is surprising because the binding interface of the complete covNHR with its complementary CHR region is much larger and can bind long CHR peptides that also include the HP binding motif, such as C34, with sub-pM affinity [22], whereas the affinity of covNHR-N-dSS for its complementary CHR region, encompassing only the MP and NTP, is only in the high nM range. This suggests that not every CHR binding motif is equally accessible to covNHR for inhibition. It has also been reported that inhibition potency of 5-helix constructs is kinetically restricted by the rate of association of the inhibitor to its target in CHR, which is only transiently exposed during fusion [38,39]. However, all NHR-based inhibitors used in previous studies contain the HP and interact with the HP binding motif, which is engaged in a tryptophan clamp stabilizing the prefusion Env conformation, and therefore, needs Env activation to become accessible [25,26]. On the other hand, accessibility of covNHR-N-dSS to its targeted CHR region may be facilitated by its small size reducing steric impediments and by the high flexibility of the CHR C-terminus [40]. In fact, a recent cryo-EM structure of full-length Env localizes the connection between the CHR end and the membrane-proximal external region (MPER) at a flexible and disordered polar segment composed of residues E^654^KNEQE^659^ [41]. This segment is actually part of the binding motif of the NTP and is engaged in a water-mediated network of hydrogen bonds in the gp41 post-fusion 6HB structure [22]. Both the NTP and the NTP-binding motif are as highly preserved as the HP and HP-binding motif (Appendix A), suggesting chief importance of this interaction in gp41 function. It is, therefore, possible that this flexible segment connecting CHR and MPER constitutes the primary target of covNHR-N-dSS in the virion context. This would explain why a stabilization of the miniprotein favoring this conformationally constrained interaction acts so strongly to increase inhibition potency.

Our results suggest two different and probably complementary modes of fusion inhibition targeting gp41 CHR. First, inhibitors containing the HP, which mainly act by targeting the HP binding motif and require its release and exposure by Env activation. Second, inhibitors targeting the C-terminal part of CHR, immediately upstream of the MPER, such as the covNHR-N miniproteins described here, which due to their small size and the higher accessibility and flexibility of their target, are less sterically restricted, and therefore, can achieve potent activities with less stringent requirements in binding affinity.

Our covNHR miniproteins have advantages over peptide-based fusion inhibitors, such as T20. First, they can be produced in recombinant form by *E. coli* expression with high yields. They spontaneously fold with the correct mimetic structure without any posttranslational modification. They are monomeric, highly soluble and stable, and can be lyophilized and reconstituted in standard buffers without structure or activity loss. All these features facilitate the scaling up of production and storage. Second, as potential antivirals, due to their polypeptidic nature, such as T20, they would need to be administered by intravenous injection but, because of their folded structure and high stability, it is highly likely that they will have a higher resistance to proteolytic degradation and longer life in the bloodstream, allowing a considerable reduction of the dosage and/or the frequency of injection, which are among the main drawbacks of T20 treatment.

These results shed light on the design of new inhibitors encompassing the N-terminal subdomain of NHR gp41 traditionally less investigated, proving the potency of the stabilized gp41’s NHR mimetics and opening up new ways of inhibiting HIV-1 by engineering new modifications increasing the stability of this region, as well as by improving the already high binding affinity for its target by adding, for instance, new motifs targeting the nearby MPER region.

## 4. Materials and Methods

### 4.1. Molecular Dynamics Simulations

All-atom molecular dynamics simulations were performed using YASARA Structure (v.17.12.24) [42] with explicit solvent (TIP3P water, the solvent density was equilibrated to a final value of 0.997 g/mL) in a periodic box with a size 10 Å larger than the protein in every dimension. In order to describe long-range electrostatics, the Particle Mesh Ewald (PME) [43] method was used with a cutoff distance of 8 Å at physiological conditions (0.9% NaCl, pH 7.4), constant temperature (298 K) using a weakly-coupled Berendsen thermostat and constant pressure (1 bar). Ewald summation was used to assign amino acids charge according to their predicted side chain pK_a_ and was neutralized by adding counterions (NaCl) [44]. The AMBER14 [45] force field was used together with multiple time step integration where intra-molecular forces were calculated every 2 fs and inter-molecular forces every 2.5 fs. The structures were initially energy-minimized using first steepest descent without electrostatics to remove steric clashes and conformational stress and subsequently relaxed by steepest descent minimization and simulated annealing (time step 2 fs, atom velocities scaled down by 0.9 every 10th step) until convergence was reached, i.e., the energy improved by less than 0.05 kJ mol^−1^ per atom during 200 steps. The minimized system was slowly heated up during an equilibration phase until the target temperature and density was reached. Every system was simulated for a minimum of 50 ns and coordinates were saved every 10 ps, yielding 5000 time points for each trajectory.

### 4.2. Protein and Peptide Samples

The NHR and CHR gp41 sequences used in this work are described in Appendix A. The reference gp41 sequence was taken from the full gp160 precursor glycoprotein of the HIV-1 BRU isolate (Swiss-Prot entry sp|P03377|ENV_HV1BR). CovNHR miniproteins were computationally designed using YASARA software. The DNA encoding the protein sequences were synthesized and cloned into pET303 expression vectors (Thermo Fisher Scientific, Waltham, USA). To facilitate purification by Ni- Sepharose affinity chromatography, the protein sequences were histidine tagged at the C terminus with the sequence GGGGSHHHHHH. The covNHR proteins were produced and purified following the protocol previously described [21]. Synthetic CHR peptides, both N-acetylated and C- amidated, were acquired from Genecust (Luxembourg), with a purity >95%. Protein and peptide concentrations were determined by UV absorption measurements at 280 nm using the extinction coefficients calculated according to their respective amino acid sequences with the ExPasy ProtParam server (https://web.expasy.org/protparam/ accessed on 2 February 2022) [46].

### 4.3. Circular Dichroism

CD spectra were recorded in a Jasco J-715 spectropolarimeter (Jasco, Tokyo, Japan) equipped with a Peltier thermostatic cell holder. Measurements of the far-UV CD spectra (260–200 nm) were made with a 1-mm path-length quartz cuvette at a protein concentration of ~15 μM. Spectra were recorded at a scan rate of 100 nm/min, 1-nm step resolution, 1-s response, and 1-nm bandwidth. The resulting spectra were usually the average of five scans and the percentage of the α-helical structure was estimated from the far-UV CD spectra as described elsewhere [47]. In thermal melting experiments, the CD signal was monitored as a function of temperature at 222 nm. Each spectrum was corrected by baseline subtraction using the blank spectrum obtained with the buffer and finally, the CD signal was normalized to molar ellipticity ([θ], in deg·dmol^−1^·cm^2^). The interaction experiments with CHR peptides were carried out at a 1:2 molar ratio between the proteins and the corresponding peptide.

### 4.4. Dynamic Light Scattering

The particle sizes of the covNHR proteins were assessed by DLS measurements using a DynaPro MS-X instrument (Wyatt, Santa Barbara, CA, USA). Dynamics software (Wyatt Technology Corporation, Santa Barbara, CA, USA) was used in data collection and processing. Sets of DLS data were measured at 25 °C with an average number of 50 acquisitions and an acquisition time of 10 s.

### 4.5. Isothermal Titration Calorimetry

ITC measurements were carried out in a Microcal VP-ITC calorimeter (Malvern Instruments, Worcestershire, UK). The protein solutions were typically titrated with 25 injections of 5 μL of the peptide solution at 480 s intervals. Protein concentration in the cell was ~20 μM, while the ligands in the syringe were typically at ~300 μM. The experiments were carried out in 50 mM sodium phosphate buffer, pH 7.4. As a blank, an independent experiment with only buffer in the calorimeter’s cell was performed with the same peptide solution to determine the corresponding heats of dilution. The experimental thermograms were baseline corrected and the peaks were integrated to determine the heats produced by each ligand injection. Finally, each heat was normalized per mole of added ligand. The resulting binding isotherms were fitted using a binding model of identical and independent sites, allowing the determination of the binding constant, K_b_, the binding enthalpy, ΔH_b_, and the binding stoichiometry, n, for each interaction. From these values, the Gibbs energy and entropy of binding could be derived as ΔG_b_ = −RT·ln K_b_ and T·ΔS_b_ = ΔH_b_ − ΔG_b_. Binding heat capacities were determined from the slope of the dependences of the binding enthalpies measured at different temperatures (ranging from 10 to 25 °C).

### 4.6. Binding to HIV-1 Envelope Spikes

The capacity of the covNHR proteins to bind soluble HIV-1 envelope proteins (Env) was determined by ELISA. Briefly, 96-well ELISA plates (Maxisorp, Nunc) were coated at 4 °C overnight with various Envs (Appendix A) in 0.1 M bicarbonate buffer (pH 9.6). After saturation with 2% BSA, 0.05% Tween in PBS for 1.5 h at 25 °C, 0.01 μM of covNHR molecules, corresponding to the protein concentrations that allowed detecting optical density changes within a linear range, (100 μL diluted in 1% BSA 0.05% Tween solution) were added and incubated for 2 h at room temperature. The plate was then washed five times and covNHR binding was detected with 100 μL anti-6X His-tag antibody conjugated to horseradish peroxidase (HRP) (Abcam, Cambridge, UK) at 1/dilution incubated for 1 h at room temperature. Antibody binding was then revealed with tetramethylbenzidine (TMB) substrate buffer, the reaction was stopped with 1 M H_2_SO_4_ and optical density was read at 450 nm with a Molecular Device Plate Reader equipped with SoftMax Pro 6 program. Background binding was measured in plates without Env and subtracted from the data. The percentage of binding was calculated using the readings with wells coated with His-tagged Env incubated with PBS buffer instead of covNHR molecules as a control for 100% binding.

### 4.7. HIV-1 Inhibitory Assays

The inhibition of HIV replication was determined using the conventional TZM-bl assay measured as a function of reductions in Tat-regulated Firefly luciferase (Luc) reporter gene expression [48]. Pseudoviruses expressing different Env were tested for HIV inhibitory potential [28]. The IC_50_, the concentration (in nM) of inhibitor inducing a 50% decrease in relative luminometer units (RLU), corresponding to a 50% decrease in virus replication was calculated by non-linear regression using a sigmoidal Hill function, as implemented in Origin software (Originlab, Northampton, MA, USA).

### 4.8. Statistical Analysis

All statistical analyses were performed using the Prism 6 scientific software. Data were expressed as the mean ± SD of 3 experiments per group. An unpaired Student’s *t*-test was used to compare the differences between two experimental groups. A *p*-value of 0.05 or less was considered to be statistically significant.

## Figures and Tables

**Figure 1 ijms-23-02794-f001:**
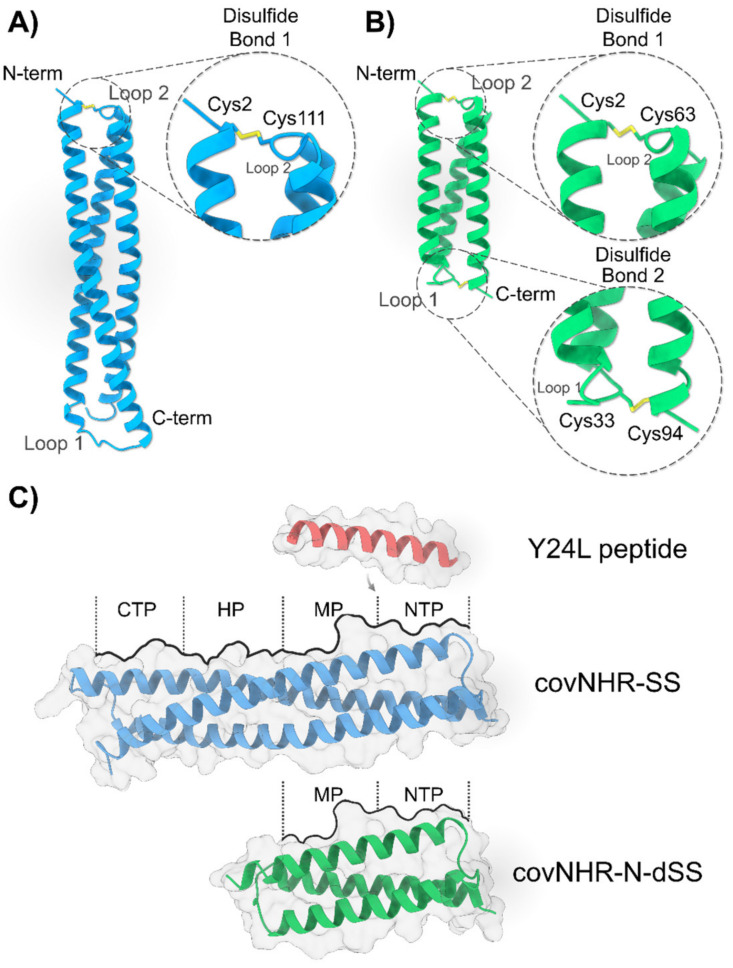
Design of covNHR miniproteins. (**A**,**B**) Ribbon models of covNHR-SS (**A**, blue) and covNHR-N-dSS (**B**, green) showing the location of the residues chosen for mutations to form disulfide bonds. Cysteines are shown in sticks and colored in yellow. (**C**) Model of the structure of the proteins and peptides involved in this study depicted in molecular surface and colored green (covNHR-N-dSS), blue (covNHR-SS) and red (Y24L peptide, gp41 residues 638–661). CovNHR is a highly accurate mimic of the full trimeric gp41 NHR coiled-coil, as such, its binding surface is also composed of four different hotspots: CTP (C-terminal Pocket), HP (Hydrophobic Pocket), MP (Middle Pocket) and NTP (N-terminal Pocket), see details in the text.

**Figure 2 ijms-23-02794-f002:**
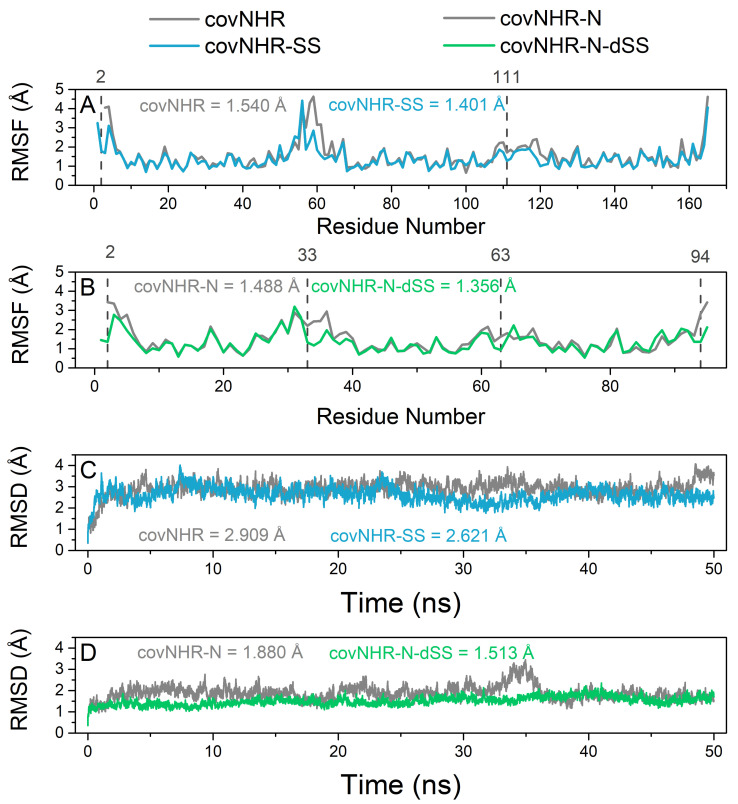
Molecular dynamics simulations analysis of the two miniproteins: covNHR-SS (blue) and covNHR-N-dSS (green) compared with their parent molecule covNHR and covNHR-N, respectively (grey). (**A**) Root mean square fluctuation (RMSF) per residue for covNHR and covNHR-SS. (**B**) RMSF per residue for covNHR-N and covNHR-N-dSS. (**C**) Evolution of mean backbone root mean square deviation (RMSD) for covNHR and covNHR-SS. (**D**) Evolution of RMSD for covNHR-N and covNHR-N-dSS. The locations of residues forming the disulfide bonds are highlighted with grey dashed lines and with their residue number shown above in A and B. Mean values for the parameters throughout the entire simulation time are also displayed in each panel with the same color code used above.

**Figure 3 ijms-23-02794-f003:**
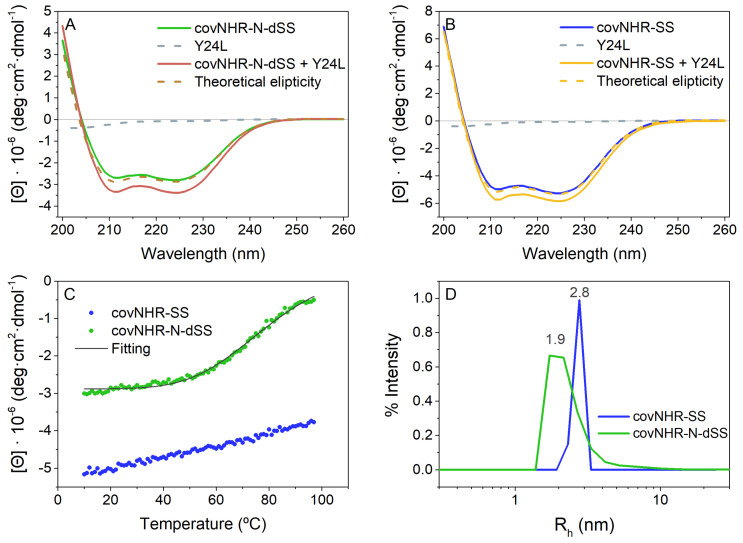
Secondary structure and thermal stability of covNHR-N-dSS (green) and covNHR-SS (blue). (**A**) Far UV CD spectra of free covNHR-N-dSS (green, solid line) and (**B**) covNHR-SS (blue, solid line) and in a (1:2) mixture with their complementary CHR peptide, Y24L (red and yellow lines, respectively). Y24L peptide alone is shown in dashed grey lines in both panels. The red and yellow dashed lines represent the theoretical sum of the spectra of the free molecules. (**C**) Thermal unfolding of covNHR-N-dSS (green symbols) and covNHR-SS (blue symbols) followed by monitoring the CD signal at 222 nm. The grey solid line corresponds to the best fitting carried out using a two-states unfolding model. (**D**) Particle size distributions measured by dynamic light scattering with solutions of covNHR-N-dSS (green) and covNHR-SS (blue). All experiments were carried out at pH 7.4 in 50 mm sodium phosphate.

**Figure 4 ijms-23-02794-f004:**
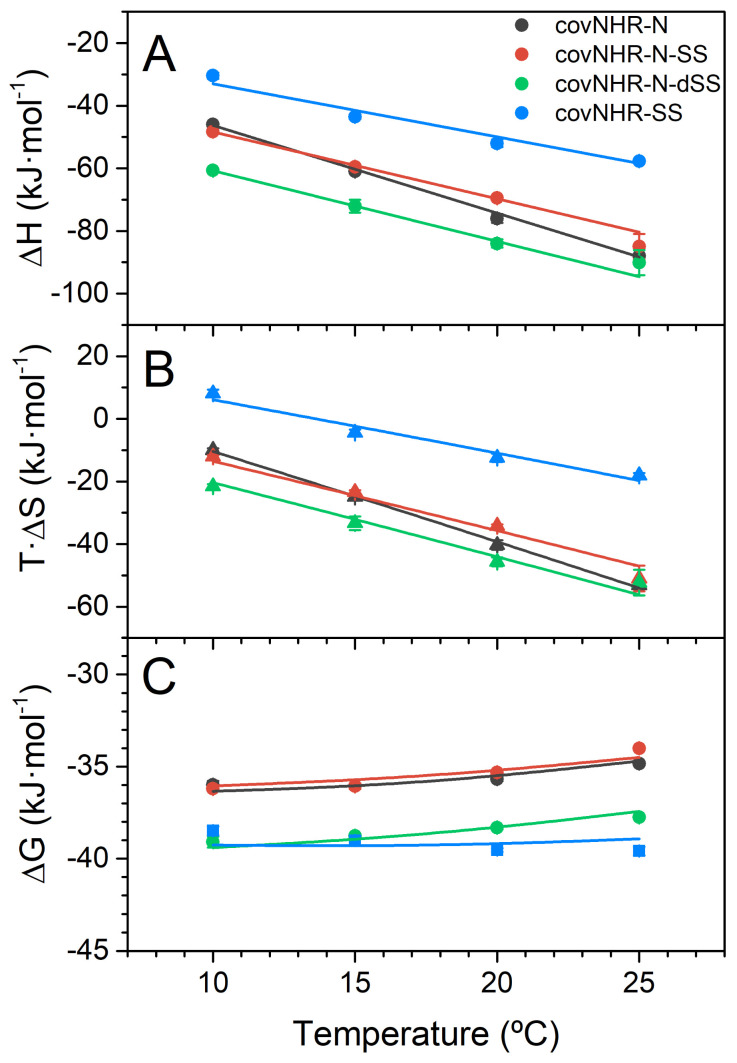
Thermodynamic parameters of Y24L peptide binding to the covNHR proteins. Data have been calculated from the parameters of Table 1 and data from [24], measured by isothermal titration calorimetry (ITC) (**A**) Binding enthalpies; (**B**) binding entropies and (**C**) binding Gibb’s energies. The symbols correspond to the values derived from experimental data and the lines represent the temperature dependencies of each parameter according to the binding heat capacity changes.

**Figure 5 ijms-23-02794-f005:**
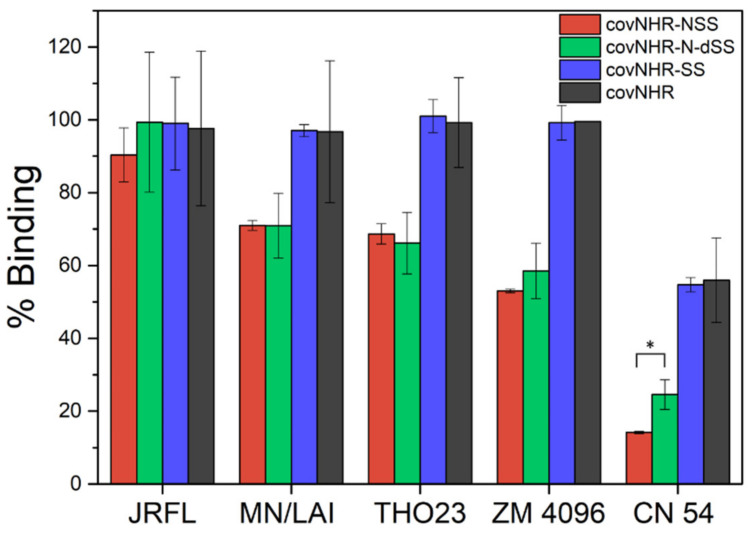
Binding of covNHR miniproteins to different soluble Envs (Appendix A) measured by ELISA. Miniproteins binding to Env was detected using anti-6X Histag Ab as primary antibody. Background binding was measured without Env and subtracted from the data; 100% positive control was measured with wells directly coated with a His-tagged Env. Data correspond to mean ± S.D. values of three independent measurements. * *p* < 0.05, difference between covNHR-N-dSS and covNHR-N-SS.

**Figure 6 ijms-23-02794-f006:**
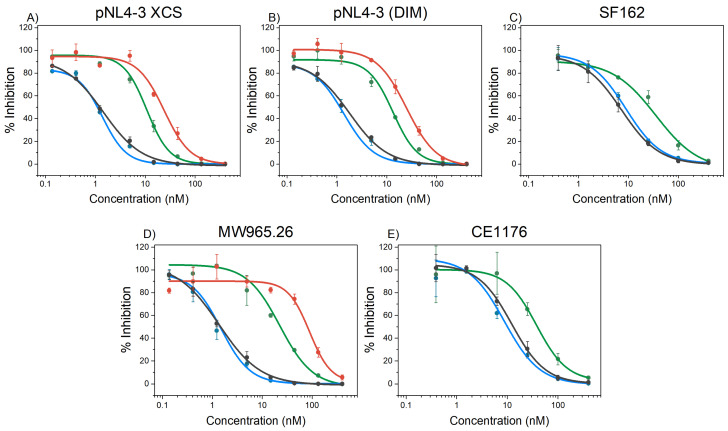
HIV-1 inhibitory activity of the miniproteins implemented in this study on different HIV-1 strains. In vitro inhibition of different HIV-1 strains infection of TZM-bL cells by fusion inhibitors, covNHR (black), covNHR-SS (blue), covNHR-N-SS (red) and covNHR-N-dSS (green), added at different concentrations. The different HIV-1 strains are (**A**) pNL4-3 XCS (pseudovirus designed for resistance to T20); (**B**) pNL4-3 (DIM) (pseudovirus designed for resistance to T20); (**C**) SF162 (pseudovirus strain); (**D**) MW965.26 (pseudovirus strain) and (**E**) CE1176 (primary isolate). Data are the mean ± S.D. of three independent measurements. Continuous lines correspond to non-linear regression curves using a sigmoidal Hill function as implemented in Origin software (Originlab, Northampton, MA, USA).

**Table 1 ijms-23-02794-t001:** Thermodynamic parameters of binding of gp41 CHR peptide Y24L to covNHR miniproteins measured by ITC.

Protein	Peptide	Temperature (°C)	K_d_ (nM)	ΔH_b_ (kJ·mol^−1^)	n	ΔC_pb_ (kJ·K^−1^·mol^−1^)
covNHR-SS	Y24L	10	80 ± 8	−30.4 ± 1.0	0.75	−1.70 ± 0.21
		15	85 ± 9	−43.5 ± 0.8	0.77	
		20	91 ± 4	−52.0 ± 1.2	0.8	
		25	116 ± 11	−57.7 ± 0.6	0.83	
covNHR-N-dSS	Y24L	10	61 ± 8	−60.7 ± 0.4	0.76	−2.26 ± 0.11
		15	94.3 ± 2.4	−72.1 ± 2.1	0.77	
		20	149 ± 6	−84.0 ± 1.3	0.78	
		25	243 ± 10	−90 ± 4	0.89	

Uncertainties in the parameters correspond to standard errors of the fittings.

**Table 2 ijms-23-02794-t002:** In vitro HIV-1 inhibition by covNHR miniproteins.

Pseudovirus	covNHR-N-SS	covNHR-N-dSS	covNHR	covNHR-SS
pNL4-3 XCS ^a^	25.6 ± 2.4	11 ± 4 **	1.3 ± 0.2	1.4 ± 0.1
pNL4-3 (DIM) ^a^	24.8 ± 2.0	13 ± 1.3 **	1.6 ± 0.1	1.4 ± 0.2
SF162	n.d.	36 ± 12	8.0 ± 1.3	8.9 ± 0.9
MW965.26	96 ± 12	23 ± 3 ***	1.5 ± 0.3	2.0 ± 0.5
CE1176 ^b^	n.d	37.9 ± 1.1	10.8 ± 1.3	8.8 ± 1.6

Inhibitory activity (IC_50_ nM ± S.D. of triplicates) was measured with the standard TZM-bl assay using different pseudoviruses; ^a^ T20-resistant strains. ^b^ Primary isolate. ** *p* < 0.01; *** *p* < 0.001, differences between covNHR-N-dSS and covNHR-N-SS.

## Data Availability

The data presented in this study are available on request from the corresponding author.

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
