# Peer review of "Conformational Stabilization of Gp41-Mimetic Miniproteins Opens Up New Ways of Inhibiting HIV-1 Fusion"

_ijms, 2022, doi:10.3390/ijms23052794_

Round 1

Reviewer 1 Report

see attached

Reviewer 2 Report

  • The manuscript entitled “Conformational stabilization of gp41-mimetic miniproteins opens up new ways of inhibiting HIV-1 fusion” presents an interesting approach and relevant findings that merit its publication. Authors strategy is based on structure-guided peptide design of HIV-1 fusion inhibitors. They take advantage of previous sequences of a small region of gp41 and used as template to create a new generation of inhibitor peptides. They increase the stability and affinity to the peptides (by them called covNHR-N miniproteins), adding disulphide bonds, to then realize evaluation of the potential inhibitory effect in vitro. The most important contributions are; (1) to demonstrate that HP is not necessary to have broad inhibition and (2) to propose two different modes of inhibition of gp41-mediated fusion.

The biophysical characterization of the miniproteins is the most solid part of the manuscript, however in the part concerning functional evaluation including the binding to Env protein and inhibition of in vitro infection requires some improvement, which I describe below:

  • Most of the data in Table 2 are from a previous publication, which reduces the validity of the results. The TZM-bl assay uses pseudoviruses expressing Env genes which differ in their infectivity from batch to batch, just as the recipient cells differ in their susceptibility to infection. So, the authors should include technical duplicates to ensure standardization of the technique, and if they have done so, show the values as supplementary data.

  • In Figure 5, the binding of the miniproteins to different soluble Envs was determined using ELISA. As described in Material and Methods, the percentage of binding was calculated from the absorbance values at 450nm. The correct expression of this type of ELISA would be the IC50, as was done in Figure 6, with the IC50 values should then be used to calculate the binding percentages between the different miniproteins.

  • In both Figures 5 and 6, mean value comparisons between groups were without statistical analysis support for such comparisons.

Minor corrections:

  • Line 167: Figure 1CD-D should be change to Figure 2C-D.

  • The colors of the CovNHR-NSS and CovNHR groups are very similar, making it difficult for the reader to compare them quickly.

  • Delete a period at the end of the title of Table 2.

  • The a T20-resistant strain legend is mentioned but not used in Table 2.

  • Line 439: Remove blue lines from previous corrections.

Reviewer 3 Report

In this manuscript, the authors investigate the effect of a disulfide bridge that was designed to staple and stabilize the N-terminal end of a mini-protein analog of the gp41 N-terminal heptad repeat (covNHR-N). They find that this additional disulfide between residues 2-64 for covNHR-N greatly stabilizes covNHR-N, increases binding affinity towards the CHR target Y24L, and improves the inhibitory potency against several HIV-1 strains. The NHR region has traditionally been hydrophobic and unstable in isolation, therefore these peptides are not extensively used as inhibitory therapeutics. This study shows that it is possible to stabilize such NHR peptides by protein engineering. Additionally, the HP region of NHR was thought to be the most important determinant of binding to CHR. As the covNHR-N mini-proteins that are devoid of the HP region could neutralize HIV-1 variants at very low concentrations, the study reveals potentially new areas of the gp41 CHR that can be efficiently targeted with novel structure-based protein design. The work is neat and carried out well, and I recommend that it for publication. There are a few minor revisions that I would suggest:

  1. It was difficult to find out what the Y24L peptide is. The fact that it is the gp41 CHR peptide is only mentioned in Figure 1C, but nowhere else in the text.
  2. Another comment on nomenclature issue: As I understand, the disulfide bridge that was used in the last publication was 34-95 which was part of the construct covNHR-NSS. This is not clearly mentioned anywhere, and the reader is not expected to go back and forth between this and earlier publications. It would great to include a key for each construct name that details information about which disulfide is present.
  3. Typo: Figure 1C-D on line 167 should be Figure 2C-D
  4. Line 198: “covNHR-N-dSS has higher negative ellipticity values than covNHR-N and covNHR-NSS”; I don’t see this on Figure 3A/B, instead covNHR-N-dSS seems to have lower ellipticity values which is only expected since they are smaller helices. Am I missing something here?
  5. Table 1 and Figure 6 show data comparison for covNHR-SS and covNHR-NdSS only, while the real comparison should be between the parent molecule covNHR-NSS and covNHR-NdSS, to test the efficacy of the disulfide. Of course, the text gives adequate information about the actual comparison, but the figures and table do not reflect on this accurately.
